

# Cellular uptake of allicin in the hCMEC/D3 human brain endothelial cells: exploring blood-brain barrier penetration in an *in vitro* model

Kankawi Satsantitham[1], Pishyaporn Sritangos[1], Sirawit Wet-osot[2], Nuannoi Chudapongse[3] and Oratai Weeranantanapan[1]

[1] School of Preclinical Sciences, Institute of Science, Suranaree University of Technology, Nakhon Ratchasima, Thailand
[2] Department of Medical Sciences, Ministry of Public Health, Nonthaburi, Thailand
[3] Department of Basic Medical Sciences, Siam University, Bangkok, Thailand

## ABSTRACT

**Background**. Allicin, a bioactive compound derived from garlic (*Allium sativum*), demonstrates antibacterial activity against a broad spectrum of bacteria including the most common meningitis pathogens. In order to advocate for allicin as a potential therapeutic candidate for bacterial meningitis, the present study aimed to assess the ability of allicin to cross the blood-brain barrier (BBB) using an *in vitro* model.

**Methods**. The cell viability of the human brain endothelial cell line hCMEC/D3 after incubation with various concentrations of allicin was investigated using an MTT assay at 3 and 24 h. Additionally, reactive oxygen species (ROS) production of allicin-treated hCMEC/D3 cells was examined at 3 h. The concentrations of allicin that were not toxic to the cells, as determined by the MTT assay, and did not significantly increase ROS generation, were then used to investigate allicin's ability to traverse the *in vitro* BBB model for 3 h. High-performance liquid chromatography (HPLC) analysis was utilized to examine the allicin concentration capable of passing the *in vitro* BBB model. The cellular uptake experiments were subsequently performed to observe the uptake of allicin into hCMEC/D3 cells. The pkCSM online tool was used to predict the absorption, distribution, metabolism, excretion, and pharmacokinetic properties of allicin and S-allylmercaptoglutathione (GSSA).

**Results**. The results from MTT assay indicated that the highest non-toxicity concentration of allicin on hCMEC/D3 cells was 5 $\mu$g/ml at 3 h and 2 $\mu$g/ml at 24 h. Allicin significantly enhanced ROS production of hCMEC/D3 cells at 10 $\mu$g/ml at 3 h. After applying the non-toxicity concentrations of allicin (0.5–5 $\mu$g/ml) to the *in vitro* BBB model for 3 h, allicin was not detectable in both apical and basolateral chambers in the presence of hCMEC/D3 cells. On the contrary, allicin was detected in both chambers in the absence of the cells. The results from cellular uptake experiments at 3 h revealed that hCMEC/D3 cells at $1 \times 10^4$ cells could uptake allicin at concentrations of 0.5, 1, and 2 $\mu$g/ml. Moreover, allicin uptake of hCMEC/D3 cells was proportional to the cell number, and the cells at $5 \times 10^4$ could completely uptake allicin at a concentration of 5 $\mu$g/ml within 0.5 h. The topological polar surface area (TPSA) predicting for allicin was determined to be 62.082 Å$^2$, indicating its potential ability to cross the BBB.

Corresponding author
Oratai Weeranantanapan,
oratai@g.sut.ac.th

Additionally, the calculated logBB value surpassing 0.3 suggests that the compound may exhibit ease of penetration through the BBB.

**Conclusion**. The present results suggested that allicin was rapidly taken up by hCMEC/D3 cells *in vitro* BBB model. The prediction results of allicin's distribution patterns suggested that the compound possesses the capability to enter the brain.

## INTRODUCTION

Allicin, a small lipophilic molecule prominently present in freshly crushed garlic (*Allium sativum*), serve as one of its primary bioactive constituents and imparts the characteristic pungent odor to garlic (*Borlinghaus et al., 2014*). Allicin possesses diverse biological properties, including the reduction of reduce blood lipid (*Abramovitz et al., 1999*), lower blood pressure (*Ried et al., 2008*), and manifestation of anti-tumor activities which would be useful for cancer therapy and prevention (*Hirsch et al., 2000*). Moreover, allicin also exhibits antimicrobial efficacy against a broad spectrum of microorganisms, encompassing bacteria, fungi, protozoa, and viruses (*Curtis et al., 2004*; *Weber et al., 1992*). Moreover, it has been reported from both *in vitro* and *in vivo* studies that allicin has antioxidative stress properties (*Zhang et al., 2017*; *El-Sheakh et al., 2016*).

Oxidative stress can result from the excessive generation or accumulation of free radical reactive species, such as the reactive oxygen species (ROS), reactive nitrogen species (RNS) and reactive sulfur species (RSS). It has been reported that accumulation of ROS in endothelial cells causes dysfunction of endothelial cells (*Shaito et al., 2022*). The study in IPEC-J2 cells, intestinal porcine enterocytes, demonstrated that allicin improves intestinal epithelial barrier function and low concentrations of allicin promoted anti-oxidative activity (*Gao et al., 2022*). Regarding therapeutic potential of allicin, the study focused on its application in treating neurodegenerative diseases, conducted using Alzheimer's transgenic mouse model exhibits that allicin mitigates cognitive impairment *via* suppressing oxidative stress (*Zhang et al., 2018*).

Glutathione (GSH), a predominant antioxidant found in the cytosol of many organisms, plays a crucial role in cellular defence against oxidative stress (*Borlinghaus et al., 2021*). Allicin has been shown to interact directly with reduced GSH, forming GSSA as a reaction product (*Rabinkov et al., 2000*). This interaction involves allicin penetrating phospholipid bilayers, such as those of vesicle-enclosed GSH and human red blood cells, allowing it to react with the thiol (-SH) groups of GSH. Remarkably, this conversion to GSSA occurs without causing membrane leakage, fusion, or aggregation, highlighting allicin's ability to permeate cellular membranes effectively while preserving their structural integrity (*Miron et al., 2000*).

The blood–brain barrier (BBB) is a complex structural framework comprising various types of cells such as pericytes, astrocytes, and brain endothelial cells interconnected by
tight junctions, thus meticulously regulating BBB permeability. Functionally, the BBB preserves the central nervous system (CNS) homeostasis by strictly controlling the passage of substances from the circulation into the brain. It concurrently acts as a protective barrier that shields the CNS from neurotoxic substances and pathogens (*Abbott et al., 2010*). Due to its highly selective permeability, BBB poses challenges for drug delivery into the CNS, thereby necessitating the development of methodologies for effective drug transport (*Larsen, Martin & Byrne, 2014*). Various types of BBB models have been used to study drug delivery to CNS for the treatment of brain disorders. Among the diverse BBB models, Transwell BBB models are the simplest *in vitro* BBB model that provides easy handling and cost-effectiveness (*Jackson et al., 2019*) (Portions of this text were previously published as part of a thesis: *Satsantitham, 2021*). The human cerebral microvascular endothelial cell line hCMEC/D3 is well-characterized and the most widely used for constructing the *in vitro* BBB models. The hCMEC/D3 cells represent a stable and easily grown BBB model, rendering it suitable for drug uptake studies (*Weksler, Romero & Couraud, 2013*).

Although the BBB can prevent the pathogenic invasion into the brain, certain bacteria possess the ability to penetrate the BBB, leading to meningitis, a serious and life-threatening disease (*Doran et al., 2016*). One of the most common pathogens that causes bacterial meningitis worldwide, with high morbidity and mortality, is *Neisseria meningitidis* (*Rouphael & Stephens, 2012*). According to our previous study, allicin exhibited antibacterial activity against *N. meningitidis* with low MIC (minimum inhibitory concentration) and MBC (minimum bactericidal concentration) values (*Satsantitham et al., 2022*). Considering the potential for allicin to traverse the BBB, this finding suggests its potential utility in the therapy of *N. meningitidis*-induced meningitis. Previous studies have reported that allicin could cross the BBB by investigating its BBB penetration properties using the SwissADME sever and showed its neuroprotective effects on ischemia-reperfusion brain injury (IRBI) *in vivo* (*Itepu et al., 2019*; *Kong et al., 2017*). However, direct evidence conclusively demonstrating allicin's BBB permeability remains lacking. Therefore, the present study aimed to evaluate allicin's capability to pass BBB using the *in vitro* model and *in silico* study.

## MATERIALS & METHODS

### Reagents

Allicin (purity >98%) and DCFDA cellular ROS detection assay kit ab113851 were purchased from Abcam (USA). MTT 3-(4,5-dimethylthiazol-2-yl)-2,5-diphenyltetrazolium bromide was obtained from Invitrogen (Eugene, OR, USA). Lucifer yellow CH dipotassium salt was acquired from Sigma (St. Louis, MO, USA).

### Cell culture

The hCMEC/D3, which is the brain microvascular endothelial cell line of the human blood–brain barrier, was purchased from Merck Millipore, MO, USA. The cells were grown in the endothelial basal medium-2 (EBM-2) containing EGM-2 MV SingleQuots® supplements and growth factors on a collagen-coated T-75 flask. Cell culture medium, supplements, and growth factors were bought from Lonza (Walkersville, MD, USA). The

collagen Type I, Rat tail (extracted from rat tail tendons) was obtained from EMD Millipore (Billerica, MA, USA). The cultured cells were constantly incubated at 37 °C in a humidified incubator with 5% $CO_2$.

## Cell viability assay

Cell viability of hCMEC/D3 cells was examined using an MTT assay. The yellow dye MTT is reduced to purple formazan form by the action of mitochondrial succinate dehydrogenase enzymes in living cells. The quantity of formazan is proportional to the number of viable cells. The hCMEC/D3 cells were seeded onto collagen-coated 96-well plates at a density of $1 \times 10^4$ cells/well for 24 h. Subsequently, the culture medium was aspirated, and the cells were treated with 100 µl of allicin at concentrations ranging from 0 to 10 µg/ml for duration of 3 and 24 h. After allicin removal, 10 µl of MTT reagent (5 mg/ml) and 100 µl of phosphate-buffered saline (PBS) were added to each well and incubated for 4 h. After MTT removal, 50 µl of dimethyl sulfoxide (DMSO) was introduced to solubilize the formazan crystals. The optical density (OD) was measured to determine the cell viability at a wavelength of 540 nm using the Multiskan Go microplate spectrophotometer (Thermo Scientific, Finland). All experiments were conducted in triplicate.

## ROS detection assay

Intracellular reactive oxygen species (ROS) formation was determined using the DCFDA cellular ROS detection assay kit following the manufacturer's protocols. Briefly, after the hCMEC/D3 cells at a density of $1.5 \times 10^4$ cells/well were allowed to adhere in a dark, clear bottom 96-well microplate with collagen-coated for 24 h. Following a single wash with PBS, the cells were stained by adding 100 µl of 15 µM fluorogenic dye DCFDA (2′,7′-dichlorofluorescin diacetate) into each well and incubated in darkness for 45 min at 37 °C. Post-DCFDA removal, the cells were subsequently washed with PBS, followed by incubation with 100 µl/well of various concentrations of allicin (ranging from 0 to 10 µg/ml) and 50 µM *tert*-Butyl hydrogen peroxide (TBHP: positive control) for 3 h. The fluorescence intensity was measured to quantify the generated cellular ROS using a fluorescence microplate reader at excitation and emission wavelengths of 485 and 535 nm, respectively, employing a multi-mode plate reader (CLARIOstar Plus, Ortenberg, Germany). The experiments were performed in triplicate.

## The construction of the *in vitro* BBB model

An *in vitro* model of BBB was generated utilizing a Transwell system that consists of the upper (apical or AP) and lower (basolateral or BL) chambers, designed to simulate the blood and brain side of the BBB, respectively. The two chambers of the Transwell insert are separated by a porous membrane (0.4 µm). Initially, the 24-well inserts (Millicell, Germany) were coated with collagen 1:20 in Dulbecco's phosphate-buffered saline (DPBS) at 200 µl/well and incubated for 1 h at 37 °C and 5% $CO_2$. After collagen removal, the hCMEC/D3 cells were seeded onto the collagen-coated membrane of the insert at a density of $1 \times 10^4$ cells/well. The cell-free insert with collagen coating served as a control insert. Subsequently, the apical and basolateral chambers of the BBB model were filled with the EBM-2 medium at 200 µl and 1,000 µl, respectively. The cells were cultured

for approximately 21 days at 37 °C and 5% $CO_2$. The culture medium in both AP and BL chambers was carefully changed every 3 days from 1X to 0.5X and 0.25X during the culture period, where the value of X refers to the concentration of the growth factors and supplements that were added into the culture medium. On day 21, the integrity of the *in vitro* BBB model was verified by measuring trans-endothelial electrical resistance (TEER) across the cell monolayers using a Millicell ERS voltohmmeter (Millipore). The integrity of the cell monolayers was also further confirmed by monitoring the passage of Lucifer yellow (LY), a paracellular marker, across the *in vitro* BBB model. For the LY permeability assay, all media was removed from the AP and BL chambers of the inserts. Then, 200 µl of LY (20 µM) and 1,000 µl of Hanks' Balanced Salt Solution (HBSS) were added into AP and BL chambers, respectively. After incubation for 1 h at 37 °C and 5% $CO_2$, the samples were collected. The concentration of LY from the BL chamber of the inserts was measured using a fluorescence microplate reader with excitation at 485 nm and emission at 530 nm. The %LY rejection of the *in vitro* BBB model was calculated using the following formula (*Nkabinde et al., 2012*).

$$\%LY \text{ rejection} = [1 - (LY_{BL}/LY_I)] \times 100$$

Where $LY_{BL}$ is the concentration of Lucifer yellow passing into the BL chamber and $LY_I$ is the initial concentration of Lucifer yellow.

## Determination of allicin in the *in vitro* BBB model by HPLC analysis

According to the cell viability assay and ROS detection results, the non-toxicity concentrations of allicin (0.5, 1, 2, and 5 µg/ml) were selected. After verifying the *in vitro* BBB model by TEER measurement and Lucifer yellow assay, the allicin test was carried out by adding 200 µl of the selected concentrations of allicin and 1,000 µl of HBSS into the AP and BL chambers of the inserts, respectively. Simultaneously, a cell-free control insert with collagen-coated was also treated with 200 µl of allicin 5 µg/ml. After incubation for 3 h at 37 °C and 5% CO2, the samples were collected. The LY assay and TEER measurement were performed again, respectively to confirm the intactness of the *in vitro* BBB model at the end of the allicin test. The concentrations of allicin that can cross the brain endothelial layer from AP to BL chamber were analyzed by high-performance liquid chromatography (HPLC) analysis using Aligent HPLC 1260 (Aligent Technologies, Santa Clara, CA, USA). The HPLC system consisted of the following components: a pump, an injector, a Hypersil ODS column (250 × 4.0 mm, 5 µm particle size), and a UV detector (254 nm). Isocratic mode operation at a flow rate of 0.5 ml/min was employed with mobile phases composed deionized $H_2O$ and methanol in a ratio of 50:50.

## Cellular uptake experiments

In explore the allicin uptake capability of hCMEC/D3 cells, cellular uptake experiments were performed. The experiments were first consisted of two conditions: (1) with cells and (2) without cells. In the first condition (with cells), the hCMEC/D3 cells were seeded in the collagen-coated wells of the 24-well plate at a density of $1 \times 10^4$ cells/well. Simultaneously, one well was also seeded with hCMEC/D3 cells at a density of $5 \times 10^4$ cells/well to

investigate whether an increases in the cell number influences allicin uptake. The second condition (without cells) was filled with a cell-free medium. All wells were filled with 500 μl of culture medium and incubated for 24 h at 37 °C and 5% $CO_2$. Afterwards, the culture medium was removed from the plate. Then, 250 μl of the non-toxicity concentrations of allicin, which are 0.5, 1, 2, and 5 μg/ml, were added into the wells with cells ($1 \times 10^4$ cells/well) and those without cells. The well with a cell density of $5 \times 10^4$ cells/well was treated with only 250 μl of 5 μg/ml allicin. After 3 hour-incubation at 37 °C and 5% $CO_2$, the supernatants were collected and subjected to HPLC to determine the allicin concentration.

To further assess whether the amount of allicin taken up by hCMEC/D3 cells exhibited proportionality to the cell number, the cells were seeded in the collagen-coated wells of the 24-well plate at a density of 0, $1 \times 10^4$, $2.5 \times 10^4$, $5 \times 10^4$, and $7.5 \times 10^4$ cells/well. After allowing the cells were allowed to attach for 24 h at 37 °C and 5% $CO_2$, the culture medium was replaced with 250 μl of 5 μg/ml allicin, followed by an incubation period at 37 °C and 5% $CO_2$ for 3 h. Then, the allicin concentration in the collected supernatants was determined by HPLC analysis.

To investigate the temporal dynamics of allicin uptake by hCMEC/D3 cells, the cells were seeded in the collagen-coated 24-well plate at a density of $5 \times 10^4$ cells/well and incubated for 24 h at 37 °C and 5% $CO_2$. Subsequently, both the well with cells and the cell-free well (without cells) were treated with 250 μl of 5 μg/ml allicin at 37 °C and 5% $CO_2$. Supernatants were collected at 0.5, 1, 1.5, 2, 2.5, and 3 h of incubation period. The allicin concentration in each collected supernatant was quantified by HPLC analysis.

### *In silico* ADME profiling predictions

Physiochemical characterization of absorption, distribution, metabolism, and excretion (ADME) properties was conducted through *in silico* analysis using the pkCSM online tool (https://biosig.lab.uq.edu.au/pkcsm/; accessed May 10, 2022).

### Statistical analysis

All experiments were performed at least in triplicate per treatment condition. The mean value is averaged of 3 individual experiments. The data presented in this study were tested for normality test and were normally distributed. All data were expressed as the mean ± standard error of the mean (SEM). Cell viability data and ROS detection data were subjected to multiple group comparisons using a one-way analysis of variance (one-way ANOVA) followed by Tukey's *post hoc* test. The significant difference between the allicin concentration in the presence and absence of hCMEC/D3 cells was assessed using an independent $t$-test. Statistical analyses were performed using the SPSS version 26 software, and GraphPad Prism software, version 10.2.3 (403). A value of $p < 0.05$ was considered statistically significant.

## RESULTS

### Effects of allicin on the cell viability and ROS formation in hCMEC/D3 cells

To investigate allicin's ability to penetrate the *in vitro* BBB model, the non-toxic concentrations of allicin were firstly determined on the hCMEC/D3 cells using the MTT assay. Figure 1A illustrates the results obtained after incubation of allicin with hCMEC/D3 cells for 3 h, revealing a significant reduction in cell viability at a concentration of 10 µg/ml compared to the untreated control group. Furthermore, results at 24 h demonstrated that allicin at the concentrations of 5 µg/ml exhibited significant decreases in cell viability (Fig. 1B). Additionally, morphological changes of the cells throughout the experimental process were shown in Figs. 1C–1D. At the toxicity concentrations of allicin, the cells were found rounding, cytoplasmic shrinking and cell death.

The formation of intracellular ROS within hCMEC/D3 cells after 3 h of allicin exposure was also investigated since the elevated ROS production can potentially overwhelm the cellular antioxidant capacity and lead to cell damage, which would be affected the *in vitro* BBB integrity. As shown in Fig. 2, the results revealed that allicin at a concentration of 10 µg/ml significantly increased the ROS level compared to the untreated control group. These findings suggest that allicin may be involved in inducing ROS-mediated damage in hCMEC/D3 cells. Moreover, these observations align with the trends observed in the cell viability assessments. Therefore, the concentrations of allicin (0.5, 1, 2, and 5 µg/ml), which were not significantly toxic to the cells, and did not significantly increase ROS generation, were selected to further investigate the ability of allicin to cross the *in vitro* BBB model.

### The ability of allicin to cross the *in vitro* BBB model

The integrity of *in vitro* BBB models was verified on day 21 by TEER measurement and LY permeability assay. The %LY rejections before allicin testing were $97.61 \pm 0.57\%$ and those after the tests were $95.04 \pm 2.26\%$. The non-toxicity concentrations of allicin (ranging from 0.5 to 5 µg/ml) were then employed to evaluate the ability of allicin to cross the *in vitro* BBB models over a 3-hour period. Surprisingly, the results from HPLC analysis of the samples from both AP and BL chambers of the *in vitro* BBB model revealed the absence of the allicin across all tested concentrations of 0.5, 1, 2, and 5 µg/ml (Figs. 3A–3H). The outcome was contrary to expectations. It was anticipated that if allicin failed to penetrate the *in vitro* BBB model, the allicin peak would be present in the AP samples due to a higher allicin concentration compared to the BL samples. However, this inconsistency led us to hypothesize that allicin degradation might occur during the experimental process. To test this hypothesis, allicin was added in the cell-free insert as a control. As illustrated in Figs. 3I–3J and Fig. S1, the HPLC results from the cell-free control insert tested with allicin 5 µg/ml demonstrated that the allicin peak was detected in both AP and BL samples, and with a retention time of approximately 8.8 min, as indicated by arrows. These findings revealed that the disappearance of allicin occurred only in the presence of hCMEC/D3 cells in the *in vitro* BBB model. Therefore, we hypothesized that allicin could be taken up by hCMEC/D3 cells.

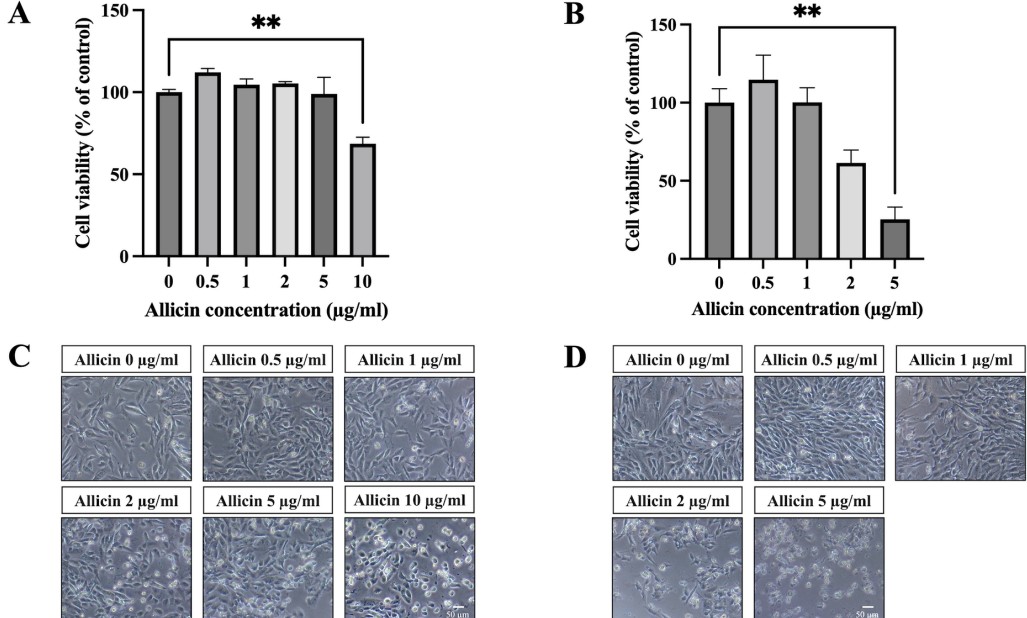

**Figure 1** **The cytotoxicity of allicin on hCMEC/D3 cells.** The cells were treated with allicin at concentrations of 0-10 μg/ml for 3 h and concentrations of 0–5 μg/ml for 24 h. Then, the cell viability was measured by MTT assay. (A–B) The percentage of cell viability compared with the untreated control at 3 and 24 h, respectively. (C–D) Morphological changes of the allicin-treated hCMEC/D3 cells at 3 and 24 h, respectively. Data are expressed as mean ± SEM ($n = 3$) and analyzed by one-way ANOVA followed by Tukey's *post hoc* test. Statistical significance was established as *$p < 0.05$ compared with the untreated control; **$p < 0.005$ compared with the untreated control. This figure was independently created by the author using original data and does not rely on any external sources.

## The uptake of allicin by hCMEC/D3 cells

Cellular uptake experiments were performed to indirectly ascertain the potential uptake of allicin by hCMEC/D3 cells within the *in vitro* BBB model. Firstly, the non-toxicity concentrations of allicin (0.5–5 μg/ml) were applied to hCMEC/D3 cells at a density of $1 \times 10^4$ cells/well and their cell-free wells for 3 h. The results showed that the allicin at concentrations of 0.5, 1, and 2 μg/ml were significantly reduced in the presence of hCMEC/D3 cells when compared to its cell-free wells. However, there was no statistical difference observed in the concentration of allicin at 5 μg/ml in the presence or absence of hCMEC/D3 cells (Fig. 4A). Thus, we postulated that the cell count utilized in this experiment might not be sufficient to uptake allicin at 5 μg/ml. The cell number used in both cellular uptake experiments and the *in vitro* BBB model remained the same at $1 \times 10^4$ cells, although the cell culture duration of the *in vitro* BBB model (21 days) was longer than the cellular uptake experiments (1 day). Hence, the number of cells that we initially used was presumably much lower than that in the *in vitro* BBB model. The number of hCMEC/D3 cells was increased to test the hypothesis that the greater number of cells, the more uptake of allicin. As shown in Fig. 4B, the HPLC results also showed that the concentration of allicin at 5 μg/ml was significantly reduced in the presence of hCMEC/D3 cells at a density of $5 \times 10^4$ cells/well compared to its cell-free well, suggesting that allicin was taken up by

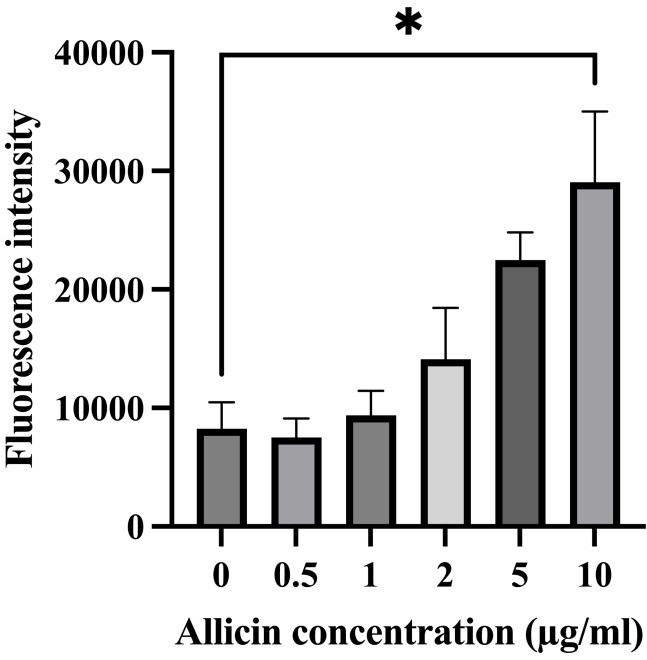

**Figure 2   Effect of allicin on ROS generation of hCMEC/D3 cells.** The cells were stained with 15 $\mu$M DCFDA for 45 min followed by incubation with allicin at concentrations of 0–10 $\mu$g/ml for 3 h. ROS was measured by ROS detection assay kit. The fluorescence intensity of cells was measured with a fluorescence microplate reader. Data are expressed as mean $\pm$ SEM ($n = 3$) and analysed by one-way ANOVA followed by Tukey's multiple comparisons test. Statistical significance was established as $^{\star}p < 0.05$ compared with the untreated control.

the hCMEC/D3 cells. As shown in Fig. 4C, the data further confirmed that the allicin uptake in hCMEC/D3 cells was proportional to the cell number as the concentration of allicin (5 $\mu$g/ml) was reduced following the increasing cell number. Allicin was completely taken up by hCMEC/D3 cells at a density of $5 \times 10^4$ cells/well. To explore whether the allicin uptake of hCMEC/D3 cells was increased through the incubation period, the concentration of allicin was quantified every 30 min until the end of experiments at 3 h. The results showed that allicin was completely uptaken by hCMEC/D3 cells ($5 \times 10^4$ cells/well) at 0.5 h, while the concentration of allicin constantly remained the same throughout the experiments in the absence of hCMEC/D3 cells (Fig. 4D). This data indicated that allicin was rapidly taken up by the cells. Taken together, the results indicated cellular uptake of allicin by hCMEC/D3 cells.

## Computational pharmacokinetic profile of allicin and its metabolite (GSSA)

The absorption, distribution, metabolism, and excretion parameters of both allicin and S-Allylmercaptoglutathione (GSSA) was presented in Table 1, with assessments conducted through the implementation of the pkCSM online software (*Pires, Blundell & Ascher, 2015*). In consideration of allicin molecular, the topological polar surface area (TPSA) was found to be below 140 Å$^2$, facilitating cellular permeation. Furthermore, it is

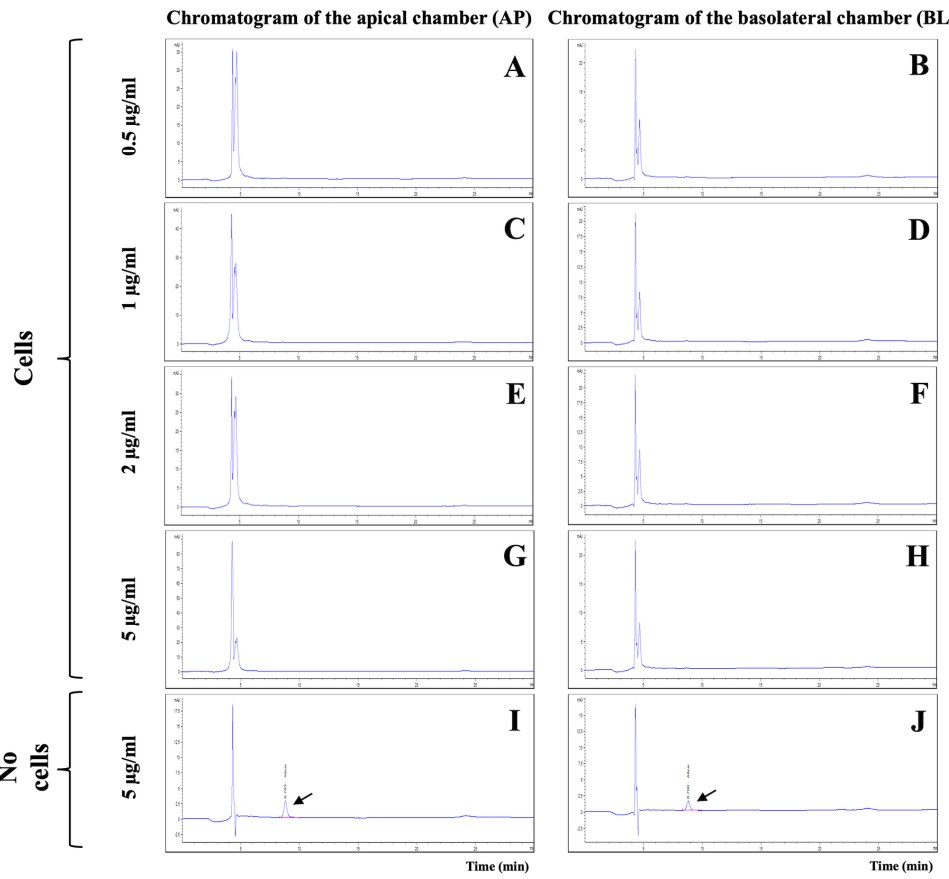

**Figure 3** **HPLC analysis of allicin samples from the *in vitro* BBB model.** Chromatogram of AP (apical chamber) and BL (basolateral chamber) samples tested with allicin (A–B) 0.5, (C–D) 1, (E–F) 2, (G–H) 5 μg/ml, and (I–J) 5 μg/ml without hCMEC/D3 cells.

noteworthy that for molecules to cross the BBB, TPSA is determined to be less than 90 Å$^2$ (*Samuel et al., 2019*).

The TPSA value of allicin was 62.082 Å$^2$, signifying its potential capability to cross the BBB. Conversely, this permeability profile was not observed for GSSA, which demonstrated a TPSA value of 146.749 Å$^2$. Indeed, these findings involve to the model assessing blood–brain barrier permeability. Moreover, a logBB value exceeding 0.3 implies that the substance may penetrate to BBB easily. The blood–brain permeability-surface area product (logPS) serves as an alternative model that excludes systemic confounding factors. Due to the logPS value of GSSA being below -3, it was determined that the compound could not penetrate the central nervous system (CNS).

Allicin exhibits high bioavailability, with absorption occurring predominantly in the small intestine, as evidenced by its intestinal absorption values exceeding 90% and Caco-2 permeability value was above 1.30. Subsequently, allicin demonstrates a decreased volume of distribution at 0.71 L/kg, underscoring its preferential distribution within the plasma compartment as opposed to various tissues. In the context of metabolism parameters, it was

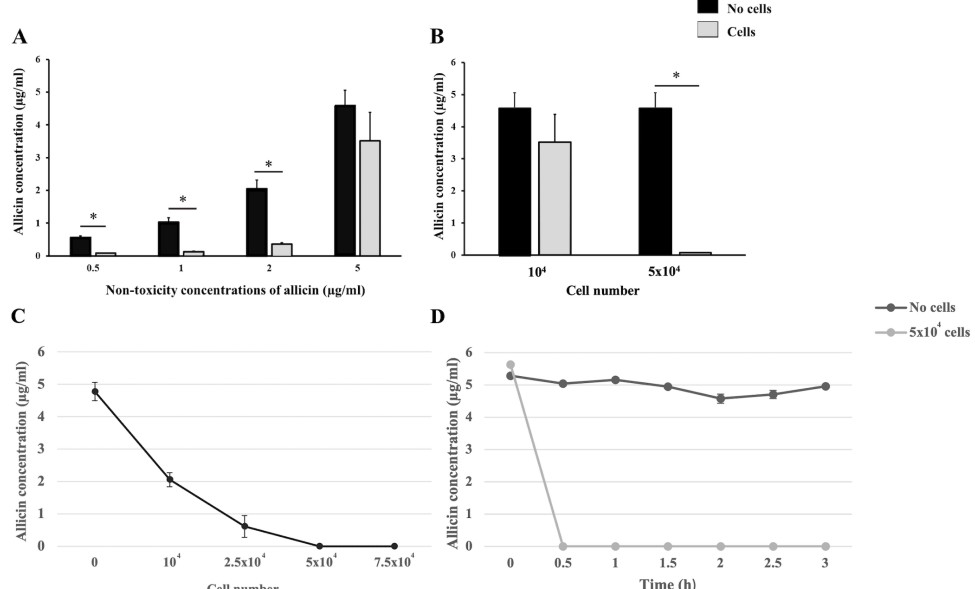

**Figure 4 The cellular uptake of allicin by hCMEC/D3 cells.** (A) Comparative concentrations of allicin at 0.5, 1, 2, and 5 μg/ml in the presence and absence of hCMEC/D3 cells at a density of $1 \times 10^4$ cells after incubation for 3 h. (B) comparative concentrations of allicin at 5 μg/ml in the presence and absence of hCMEC/D3 cells at a density of $1 \times 10^4$ and $5 \times 10^4$ cells after incubation for 3 h. The allicin concentration in each collected supernatant was determined by HPLC analysis. Data are expressed as mean ± SEM ($n = 3$) and analyzed by an independent $t$-test. Statistical significance was established as *$p < 0.05$. (C) The hCMEC/D3 cellular uptake of allicin was proportionally increased to the cell number. The cells at a density of 0, $1 \times 10^4$, $2.5 \times 10^4$, $5 \times 10^4$ and $7.5 \times 10^4$ cells were treated with allicin (5 μg/ml) for 3 h. (D) Allicin was rapidly taken up by hCMEC/D3 cells. Allicin (5 μg/ml) was tested with two conditions: "with cells" ($5 \times 10^4$ cells) and "without cells". Then the supernatants were collected at 0, 0.5, 1, 1.5, 2, 2.5 and 3 h. The allicin concentration in each collected supernatant was determined by HPLC analysis. Data are expressed as mean ± SEM ($n = 3$).

observed that allicin and GSSA cannot serve as substrates for cytochromes 2D6 (CYP2D6) and 3A4 (CYP3A4). These predicting data suggest that allicin is not metabolized by the liver. Consequently, it is reasonable to infer that alternative metabolic pathways may facilitate the transportation of allicin to the brain. Both allicin and GSSA exhibited a lack of inhibitory effects on CYP2D6 and CYP3A4, indicating a lack of interference with the biotransformation processes mediated by these specific cytochrome P450 enzymes. Regarding their *in-silico* excretion profile, allicin display a faster clearance rate compared to GSSA (0.714 and 0.251 mL/min/kg).

## DISCUSSION

Among a variety of biological properties of allicin, which is a natural active compound derived from garlic, the broad-spectrum antibacterial effect of allicin could be one of the therapeutic options for bacterial meningitis. Previous studies have reported that allicin exhibits antibacterial activity against the most common pathogens of bacterial meningitis,

**Table 1  Prediction of ADME indicators for allicin and S-allylmercaptoglutathione (GSSA).**

| Property | Model name | Predicted Value | | Units |
|---|---|---|---|---|
| | | **Allicin** | **GSSA** | |
| Molecular | log Po/w[a] | 1.7553 | −0.8362 | – |
| | TPSA[b] | 62.082 | 146.749 | $Å^2$ |
| | HBA[c] | 2 | 6 | bond acceptors |
| | HBD[d] | 0 | 6 | bond donors |
| | n-rot[e] | 5 | 13 | bonds |
| Absorption | Caco-2 permeability | 1.313 | −0.401 | log Papp in $10^{-6}$ cm/s |
| | Human intestinal absorption | 96.468 | 0 | % Absorbed |
| | Skin permeability | −1.869 | −2.735 | log Kp |
| Distribution | VDss (human) | −0.041 | −1.321 | log L/Kg |
| | BBB permeability | 0.51 | −1.244 | log BB |
| | CNS permeability | −2.312 | −3.475 | log PS |
| Metabolism | CYP2D6 substrate | No | No | Yes/No |
| | CYP3A4 substrate | No | No | Yes/No |
| | CYP2D6 inhibitor | No | No | Yes/No |
| | CYP3A4 inhibitor | No | No | Yes/No |
| Excretion | Total Clearance | 0.714 | 0.251 | log mL/min/Kg |
| | Renal OCT2 substrate | No | No | Yes/No |

**Notes.**
[a] log Po/w, logarithm of partition coefficient between *n*-octanol and water.
[b] TPSA, topological polar surface area.
[c] HBA, number of hydrogen bond acceptors.
[d] HBD, number of hydrogen bond donors.
[e] n-rot., number of rotatable bonds.

such as *N. meningitidis*, *Streptococcus pneumoniae* (multi-drug resistant and non-multi-drug resistant strains), and *Listeria monocytogenes* (*Imani Rad et al., 2017*; *Reiter et al., 2017*; *Shrivastava & Garg, 2015*). The pathogenesis stage of bacterial meningitis, in which the bacterial pathogens penetrate across the BBB, increases several inflammatory mediators, including ROS, within the cerebrospinal fluid (CSF). Thus, the elevation of ROS was one of the crucial factors that induce BBB disruption and lead to meningitis progression (*Agyeman, Grandgirard & Leib, 2017*). The current study demonstrated that allicin at non-toxicity concentrations did not significantly affect the ROS level in hCMEC/D3 cells, which is consistent with the previous studies in human umbilical vein endothelial cells (HUVECs). It has been revealed that the non-toxicity doses of allicin (100 µM) and allicin in garlic juice (<0.0094 mM) on HUVEC cells did not elevate the intracellular ROS levels (*Chen et al., 2016*; *Gruhlke et al., 2016*). Although the results from *Gruhlke et al. (2016)* demonstrated that synthetic allicin caused low DCFDA fluorescence in HUVEC cells at high concentrations, whereas allicin in garlic juice slightly increased the formation of ROS. This study showed a similar trend to our results that the toxic dose of allicin, directly derived from garlic, promoted ROS generation in hCMEC/D3 cells. The non-toxicity concentrations of allicin on hCMEC/D3 cells were defined in a range of 0.5–5 µg/ml. In comparison to our previous study, the MIC and MBC values of allicin against *N. meningitidis* were 3 and 4 µg/ml, respectively (*Satsantitham et al., 2022*). Therefore, this

finding suggested that in circumstances in which allicin can pass the BBB, it may cross the BBB with sufficient doses to inhibit *N. meningitidis*, the leading cause of bacterial meningitis worldwide. Moreover, allicin was found to have a synergistic effect with chloramphenicol, one of the effective antibiotics for the treatment of *N. meningitidis*-causing meningitis. The previous study has reported that chloramphenicol was discovered to have a synergistic effect with allicin against *Mycobacterium tuberculosis* (*Gupta & Viswanathan, 1955*). Thus, if allicin is proven to have the ability to cross the BBB, it could be considered a therapeutic alternative for meningitis caused by *N. meningitidis*.

As previously mentioned, the possibilities of allicin to cross the BBB have been reported. The *in vivo* study revealed that allicin exhibits neuroprotective effects on IRBI in mice, suggesting that allicin may have the ability to pass BBB to attenuate the IRBI (*Kong et al., 2017*). Another previous study also provided evidence by constructing the 2D structure of allicin and analyzed that its polar surface area (PSA) could penetrate the BBB using SwissADME (*Itepu et al., 2019*), a finding consistent with our results from *in silico* studies using the pkCSM online tool. Although the *in silico* study provides theoretical predictions and suggestions, it does not serve as direct evidence. Under biological conditions, several factors influence the transport of chemicals across the cellular membrane. Therefore, our *in silico* results do not conclusively demonstrate allicin's ability to traverse an *in vitro* BBB model.

The HPLC analysis demonstrated that allicin was absent in samples from both the apical (AP) and basolateral (BL) chambers of the *in vitro* BBB model. We rigorously assessed the integrity of the *in vitro* BBB model both before and after allicin testing using TEER measurements and Lucifer yellow (LY) permeability assays. If the quality of the BBB model had been compromised, such as through damage to BBB integrity during allicin testing, one would expect to find allicin present in the BL chambers, due to a small molecular size of allicin. Therefore, we assert that our *in vitro* BBB model was adequately qualified to be used for investigation. Although it could not conclude that allicin can cross the *in vitro* BBB model, our present study revealed the uptake of allicin in the human brain microvascular endothelial cells, which considered as the key element of BBB formation. Altogether, our results led to the presumption that at 3 h, allicin in the AP chamber, which was supposed to pass the BBB into the BL chamber, was rapidly taken up by hCMEC/D3 cells or passed in a small amount, which is unable to be detected by HPLC analysis. Moreover, our findings from the uptake experiment exhibit rapid cellular uptake of allicin, as early as 0.5 h post-treatment (Fig. 4D). This observation aligns with the prior research demonstrating fast diffusion and permeation of allicin not only in artificial lipid bilayer membranes but also across natural cell membranes, as observed in red blood cells (*Miron et al., 2000*). Additionally, studies have shown allicin's ability to penetrate cells and readily interact with cellular thiols (*Rabinkov et al., 2000*). Considering allicin's lipophilic nature and small molecular size (162.28 Da), we hypothesized that allicin was taken up by the hCMEC/D3 cells *via* the transcellular lipophilic pathway without the necessity of using any carrier proteins.

After cellular uptake of allicin in hCMEC/D3 cells, we proposed two potential scenarios. First, allicin may undergo release after cellular uptake, facilitating its crossing of BBB

as depicted in Fig. 5A. This proposition aligns with the forecasted BBB penetration capability of allicin as reported by *Itepu et al. (2019)* using the SwissADME server and corroborates findings from our study employing pkCSM tools. The calculation from pkCSM tools revealed the potential of allicin for brain penetration, which correlated with their properties such as low polar surface areas, lack of hydrogen bond donors, and low molecular weights compared to the 25 top-selling CNS drugs in 2004 that achieved BBB penetration (*Hitchcock & Pennington, 2006*). Generally, the blood-circulating molecules can selectively pass through the BBB by several transport mechanisms, including paracellular transport, transcellular lipophilic pathway, transporter proteins, receptor-mediated endocytosis, and absorptive transcytosis (*Jena, McErlean & McCarthy, 2020*). According to the lipophilic property of allicin and our allicin uptake results, this study proposes that allicin (162.28 Da), might be able to cross the BBB *via* the transcellular route in the same manner as propranolol (259.34 Da), morphine (285.34 Da), and midazolam (325.78 Da), the small lipid-soluble molecules, which reported to pass the hCMEC/D3 monolayers *via* the transcellular lipophilic pathway (*Poller et al., 2008*). Moreover, the low molecular weight of allicin suits the criteria that lipid-soluble molecules smaller than 400 Da can cross the BBB *via* the transcellular lipophilic pathway (*Curley & Cady, 2018*). Although, there was a desire to investigate longer incubation times of allicin in the *in vitro* BBB model, a limitation of the recent study is degradation of allicin. It has been reported that thermostability of allicin, especially at low concentrations, leads to rapid degradation (*Fujisawa et al., 2008*; *Wang et al., 2015*) . The cytotoxicity results obtained at 24 h indicated that the highest nontoxicity concentration of allicin is 2 μg/ml, therefore, we are limited in our ability to definitively rule out this possibility.

Another possibility would be after the cellular uptake of allicin, it interacts with the reduced GSH and changes to GSSA (*Rabinkov et al., 2000*) (Fig. 5B). The previous study demonstrated that allicin can easily penetrate through the phospholipid bilayers of vesicle enclosed GSH and human red blood cells (containing GSH) to interact with the thiol (-SH) groups and give GSSA as the reaction product. This interaction occurred without causing membrane leakage, fusion, or aggregation (*Miron et al., 2000*). Interestingly, GSH is a major thiol compound within mammalian cells, including brain endothelial cells (*Li et al., 2012*). Another study in human endothelial cells also revealed that the content of intracellular GSH in HUVEC cells was rapidly depleted after allicin treatment (*Gruhlke et al., 2016*). Therefore, the findings from these previous studies provide another clue that allicin in the BL chamber of our *in vitro* BBB model could not be detected because it might interact with GSH in hCMEC/D3 cells and turn into GSSA.

Taken together, our findings suggest rapid cellular uptake of allicin in hCMEC/D3 cells. However, after allicin cellular uptake, our study does not differentiate between the two postulated mechanisms. Further investigation is needed to determine whether allicin traverses the BBB *via* the transcellular route or undergoes conversion to GSSA, leading to entrapment within the cells.

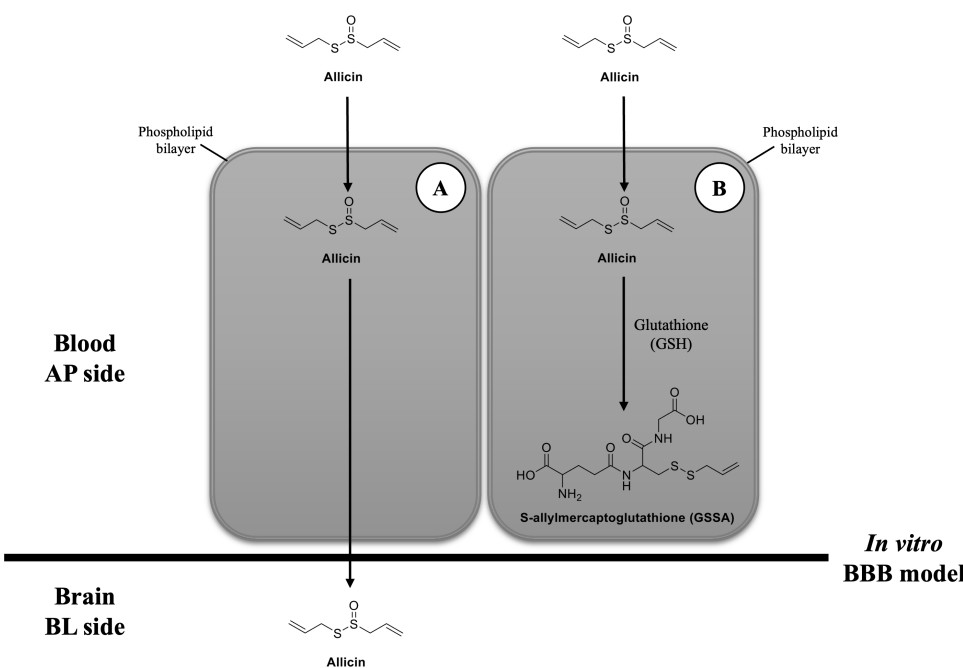

**Figure 5  Possibilities of transport mechanism of allicin by BBB.** (A) The first possibility is that the allicin traverses BBB *via* transcellular pathway after hCMEC/D3 cell permeation and (B) the second possibility is that it is taken up by hCMEC/D3 cells, reacts with GSH, changes to GSSA, and then entrapped within the cells. Original figure created by the author.

## CONCLUSIONS

While our current findings do not definitively establish allicin's capacity to traverse the *in vitro* BBB model, they significantly contribute to elucidating the mechanisms underlying its cellular uptake. The comparative distribution profiling of allicin led to the conclusion that the compound could potentially penetrate the blood–brain barrier. The significance in the context of allicin potential therapeutic application, particularly in diseases such as bacterial meningitis, is crucial for effective treatment. Moreover, understanding how allicin gets into cells could help create new ways to deliver treatments for brain diseases. In summary, our research not only helps us understand how allicin works in cells but also sets the stage for development of novel delivery strategies aimed for therapeutic neurological disorders.

## ACKNOWLEDGEMENTS

We would like to thank Miss Janjira Wongwiwattana for her assistance and advice in HPLC techniques.

### Funding

This work was financially supported by the Suranaree University of Technology Research and Development Fund (IRD1-107-67-12-24). The funders had no role in study design, data collection and analysis, decision to publish, or preparation of the manuscript.

### Grant Disclosures

The following grant information was disclosed by the authors:
Suranaree University of Technology Research and Development Fund: IRD1-107-67-12-24.

### Competing Interests

The authors declare there are no competing interests.

### Author Contributions

- Kankawi Satsantitham conceived and designed the experiments, performed the experiments, analyzed the data, prepared figures and/or tables, authored or reviewed drafts of the article, and approved the final draft.
- Pishyaporn Sritangos conceived and designed the experiments, analyzed the data, authored or reviewed drafts of the article, and approved the final draft.
- Sirawit Wet-osot conceived and designed the experiments, performed the experiments, analyzed the data, prepared figures and/or tables, authored or reviewed drafts of the article, and approved the final draft.
- Nuannoi Chudapongse conceived and designed the experiments, authored or reviewed drafts of the article, and approved the final draft.
- Oratai Weeranantanapan conceived and designed the experiments, performed the experiments, analyzed the data, prepared figures and/or tables, authored or reviewed drafts of the article, and approved the final draft.

### Data Availability

The raw data are available in the Supplementary Files.

### Supplemental Information

Supplemental information for this article can be found online at http://dx.doi.org/10.7717/peerj.17742#supplemental-information.

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
