# Peer review of "Cellular uptake of allicin in the hCMEC/D3 human brain endothelial cells: exploring blood-brain barrier penetration in an in vitro model"

_PeerJ, doi:10.7717/peerj.17742_

## Round 0.1 · original submission · Major Revisions

Thank you for submitting your manuscript to PeerJ journal, which has been through the peer-review process. Reviewer comments are below. When revising your manuscript, please carefully consider all issues mentioned in the reviewers' comments: please outline every change made in response to their comments and provide suitable rebuttals for any comments not addressed. Please note that your revised submission may need to be re-reviewed.

Reviewer 1 ·

Basic reporting

1. In the introduction, there needs to be more background information on why ROS levels of hCMEC/D3 under the Allicin treatment are investigated. There is a missing link between ROS and the potential therapeutic effects of Allicin.
2. Suggesting referred to 'Determination of allicin in the in vitro BBB model by HPLC analysis' in the supplementary figures in the result of 'The ability of allicin to cross the in vitro BBB model' for better understanding.
3. There is a lack of reference in the synergetic effects of Allicin and chloramphenicol in meningitis (Line 351-352).

Experimental design

1.In line 262, the author hypothesized Acillin degradation might occur. I assumed the degradation means the breakdown of Acillin into small molecules. However, later experiments with blank transwells tried to determine whether Acillin penetrates or is being absorbed in in vitro BBB models. A previous study indicated that Acillin degraded around 40C in an aqueous solution, and only approximately 30% was left in this condition. A small peak with a longer retention time could be observed in UPLC (Wang, HJ. (2015), J. Sci. Food Agric., 95: 1838-1844). It is reasonable to speculate that some Acillin degraded before penetration while others were uptook by the hCMEC/D3 cell layer.

Validity of the findings

1. In Figure 1C, the first graph shows that the control cells untreated with Allicin have approximately 50% cell death. A better cell culture condition needs to be determined before the Allicin treatment, even though normalization was done later. The Figure 1D looks better.
2. In Lines 403-404, the author falsely stated that 1µg/ml is below the detectable threshold of HPLC. As shown in the supplementary data, an area peak of 17 could be seen in the chromatogram of Allicin standard curve at 0.5 µg/ml. In Figure 4, Allicin concentrations in the presence of cells after incubations in 0.5, 1, 2µg/mg groups are all below 1µg/ml, which were measured through HPLC.
3. There is a lack of HPLC data on the cell-free control insert tested with allicin at other concentrations (0.5, 1, 2 µg/ml) as controls for the same concentration of Allicin in the BBB in vitro model.

Reviewer 2 ·

Basic reporting

The authors Satsantitham et al., reported the compound allicin derived from garlic is able to cross the blood brain barrier using the in-vitro bbb cell line human brain endothelial cell line hCMEC/D3, and further showed its effect on ROS production using assays and HPLC.

Clearly detailed methods and content was presented with sufficient literature. The data is well reported. The authors should report n (samples) used and how the data is averaged.

Experimental design

The design is good and within the scope. The authors used assays and HPLC to detect the compound and further, presented the data clearly. Methods and statistical analysis was done correctly.

Validity of the findings

The authors provided robust reasoning to the claims, though the work was presented in-vitro and should not be considered in-vivo blood brain barrier with the authors have clearly presented.

The findings were simple and easy to follow, well structured content.

Reviewer 3 ·

Basic reporting

In this article, the manuscript assessed the cellular uptake of allicin into the hCMEC/D3 and its effect on ROS generation in the in vitro BBB model.
The authors’ statement showed inconsistency when they concluded “The present results suggested that allicin was taken up by hCMEC/D3 cells in vitro BBB model. The prediction results of allicin's distribution patterns suggested that the compound possesses the capability to enter the brain. However, in the results section they describe “After applying the non-toxicity concentrations of allicin (0.5-5 μg/ml) to the in vitro BBB model for 3 h, allicin was not detectable in both apical and basolateral chambers in the presence of hCMEC/D3 cells.” This description was incorrect which leads to wrong mechanism of their study.

Experimental design

Material and methods: How the authors calculate n to obtain statistical power. Explain how the sample size was decided. Provide details of any a priori sample size calculation, if done.
Provide details of the statistical methods used for each analysis, including the software used
Describe any methods used to assess whether the data met the assumptions of the statistical approach, and what was done if the assumptions were not met.
The authors used different concentrations of allicin for each assay, so in the MMT assay they used 5 µ/ml, in the assay of ROS production used 10 µ/ml, but when they assessed the ability of allicin to traverse the in vitro BBB they used 0.5-5 μg/ml with no detectable concentrations in chambers. To advocate for allicin as a potential therapeutic candidate for bacterial meningitis, the concentrations of allicin must be the same between assays. In addition, as the authors report, in the presence of cells allicin was not detected in chambers suggesting that its beneficial effects can´t be warranted in BBB. From a therapeutic point of view, it is important to demonstrate that allicin can cross the BBB and exert its bactericide effects.

Validity of the findings

The authors examined the ROS production induced with the allicin treatment, but it is well known that even commercial drugs have side effects when are used in high concentrations. Therefore, it may be more interesting to use concentrations that could be found in cerebrospinal fluid to achieve therapeutic effects.
On the other hand, the therapeutic effects of allicin are well-known, and they are related to their antibiotic, antifungal, anti-inflammatory, and antioxidant properties (doi: 10.3390/antiox11010087). Allicin is hydrophobic in nature, can efficiently cross the cellular membranes, and behaves as a reactive sulfur species (RSS) inside the cells. Thus, their effects are independent of ROS production. Even, the neuroprotective effects of allicin were through the inhibition of oxidative stress, and the allicin concentration were higher 10-50 µM as used in this study (<10 µm/ml) (DOI https://doi.org/10.1039/C4FO00761A).
Additionally, the author did not explain the cellular mechanism through allicin-induced ROS. However, in the discussion section, the authors suggest that allicin can disappear to synthesize glutathione an endogenous antioxidant. This statement is contradictory to the results which showed an increase in ROS formation, please explain.

The authors claim “allicin was not detectable in both apical and basolateral chambers in the presence of hCMEC/D3 cells. On the contrary, allicin was detected in both chambers in the absence of the cells.” Which suggest that allicin is metabolized in by the cells. However, they conclude “The prediction results of allicin's distribution patterns suggested that the compound possesses the capability to enter the brain”, this statement must be demonstrated because there are no evidences that support the conclusion.

---

## Round 0.2 · Minor Revisions

The authors are suggested to address the final comment raised by Reviewer 1.

Reviewer 1 ·

Basic reporting

1. I suggest the author include the reaction of Allicin and GSH to GSSA in the background. Without this introduction, it would be hard to understand why the author conducted computational profiling of GSSA in the method.

Experimental design

No comment

Validity of the findings

No comment

Reviewer 3 ·

Basic reporting

The manuscript was reviewed carefully, included additional experiments, and was significantly improved. Also, they made the necessary adjustments in the title and conclusions according to their results.
I don't have any comments

Experimental design

No comment

Validity of the findings

No comment

Additional comments

The manuscript was reviewed carefully, included additional experiments, and was significantly improved. Also, they made the necessary adjustments in the title and conclusions according to their results.
I don't have any comments

---

## Round 0.3 · accepted · Accept

The authors have addressed all of the reviewers' comments.